# SymLearn: A Symbiotic Crowd-AI Collective Learning Framework to Web-based Healthcare Policy Adherence Assessment

## ABSTRACT

This paper develops a symbiotic human-AI collective learning framework that explores the complementary strengths of both AI and crowdsourced human intelligence to address a novel *Web-based healthcare-policy-adherence assessment (WebHA)* problem. In particular, the objective of the WebHA problem is to automatically assess people's public health policy adherence during emergent global health crisis events (e.g., COVID-19, MonkeyPox) by exploring massive social media imagery data. Recent advances in human-AI systems exhibit a significant potential in addressing the intricate imagery-based classification problems like WebHA by leveraging the collective intelligence of both humans and AI. This paper aims to address the limitation of existing human-AI systems that often rely heavily on human intelligence to improve AI model performance while overlooking the fact that humans themselves can be fallible and prone to errors. To address the above limitation, this paper develops SymLearn, a symbiotic human-AI co-learning framework that leverages human intelligence to troubleshoot and fine-tune the AI model while using AI models to guide human crowd workers to reduce the inherent human errors in their labels. Extensive experiments on two real-world WebHA applications show that SymLearn clearly outperforms the state-of-the-art baselines by improving WebHA performance and reducing crowd response delay.

## CCS CONCEPTS

• **Human-centered computing** → **Collaborative and social computing**.

## KEYWORDS

Social Media, Public Health, Crowdsourcing, Human-AI Collaboration

**ACM Reference Format:**
Anonymous Author(s). 2023. SymLearn: A Symbiotic Crowd-AI Collective Learning Framework to Web-based Healthcare Policy Adherence Assessment. In *Proceedings of ACM Conference (Conference'17)*. ACM, New York, NY, USA, 11 pages. https://doi.org/10.1145/nnnnnnn.nnnnnnn

## 1 INTRODUCTION

Web-based social media platform has emerged as a pervasive application paradigm that allows individuals and their devices to work in tandem to collaboratively report their observations about the

physical world [12]. In particular, this application paradigm has garnered significant attention for its capability to collect real-time information about public health at scale [43]. As the world faces new global health crisis (e.g., COVID-19, Polio, and MonkeyPox), public health and well-being are increasingly at risk, especially among vulnerable populations such as the elderly and immuno-compromised individuals [5]. Governments have designed various public health policies to address the prevailing health crisis and mitigate its adverse effects on the populace. Examples of such policies include mask wearing, social distancing, hand-washing and sanitization. However, to effectively implement and adjust these policies, policymakers would need the accurate and timely information on public health policy adherence [29]. Social media have surfaced as a ubiquitous Web platform for accessing an unparalleled volume of timely observations about public health practice through the imagery data posted by common citizens [43]. In this paper, we focus on a *Web-based healthcare-policy-adherence assessment (WebHA)* problem where the goal is to assess people's healthcare policy adherence by leveraging massive social media imagery data.

Recent progress has been made in addressing the WebHA problem in Web-based applications and AI communities [1, 23, 35]. Current solutions often focus on leveraging advanced AI models (e.g., transformers, CNNs) that can effectively identify visual characteristics related to public health practices, ensuring reasonable WebHA performance [49]. However, these AI models often require a large amount of high-quality training data from the studied WebHA application and can encounter undesirable errors due to the complex and noisy nature of social media images (e.g., these images can be captured by various cameras with diversified angles, resolutions, and backgrounds) [29]. Recent advances in human-AI collective systems have shown great potential to address the limitations of AI models in solving complex imagery-based classification problems such as WebHA by exploring the collective intelligence of both humans and AI [38, 52]. These systems often use AI models to analyze a vast amount of imagery data while leveraging human intelligence (HI) to troubleshoot, fine-tune, and boost the performance of AI models [48]. However, a major limitation in current human-AI collective systems is that they rely heavily on HI to improve AI model performance while overlooking the fact that humans themselves can be fallible and prone to errors [55]. Such imperfect HI could potentially collapse the AI models during the model training process and lead to suboptimal application performance [36]. To address the above challenge, current solutions apply various active learning and label aggregation methods to improve the overall accuracy of the human labels [13]. However, these approaches fail when common mistakes exist in the human labels [14].

To address the above limitations, this paper proposes a *symbiotic* human-AI co-learning framework that explores the collective power of both AI and HI. Our design is inspired by the symbiosis

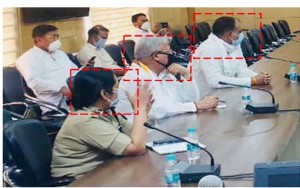 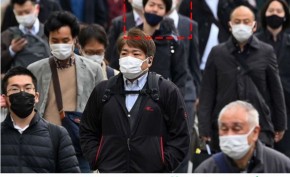

AI: Not Adhering ✗
Human: Adhering ✓
(A)

AI: Not Adhering ✓
Human: Adhering ✗
(B)

**Figure 1: Illustrations of the symbiotic relationship between humans and AI in mask wearing adherence assessment (red color: incorrect estimation; green color: correct estimation).**

from biology where two species establish a mutually beneficial relationship. For instance, flowers rely on bees to cross-pollinate their female plants while bees feed on the flowers. Similarly, the idea of our framework is to explore the symbiotic relationship between AI and HI and boost the overall performance of the integrated system for the target application. In particular, our system leverages HI to troubleshoot and fine-tune the AI model while using AI models to guide human workers to overcome the inherent limitations of HI (e.g., perceptual resolution limitations and ignorance of fine-grained details). For example, in Figure 1, we observe that the AI model can mistakenly treat the people who are not facing directly toward the camera as not adhering to the mask wearing policy as shown in (A), while humans can recognize such cases when AI model fails and identify correct WebHA labels. Meanwhile, we also observed that humans can make incorrect estimations of the WebHA label by ignoring the people in the back who are not following the mask wearing policy, as shown in (B). In this case, AI model can make an accurate estimation by analyzing the details that humans may overlook. Therefore, the AI model can guide humans to pay attention to these individuals and improve the accuracy of human label quality. However, designing such a symbiotic human-AI co-learning framework poses two technical challenges.

The first challenge is the complex interdependence between AI and HI. We observe that a "chicken-and-egg" dilemma arises when one form of intelligence relies on the reliable outputs of the other [33]. Specifically, on one hand, AI models often need accurate human workers to troubleshoot and improve their performance (e.g., Figure 1 (A)). On the other hand, human workers need precise AI feedback on the quality of their labels and reminders of the important public health practice-related visual details that they might have overlooked, thereby reducing the likelihood of incorrect annotations (e.g., Figure 1 (B)) [46]. However, neither AI nor HI is perfect as shown in Figure 1. Therefore, obtaining accurate WebHA labels in human-AI systems is challenging given such interdependence between two types of intelligence [51].

The second challenge is to optimize the trade-off between label quality and response delay of the human-AI co-learning system. One straightforward solution for establishing a human-AI co-learning system is to directly recruit public health specialists to obtain a good number of high-quality WebHA labels and then leverage those labels to troubleshoot and fine-tune the AI model [38]. However, this approach is not practical due to its high labor costs

and low efficiency, especially in the case of processing massive amounts of social media data inputs in WebHA applications [30]. To obtain readily available and cost-effective human intelligence, we can leverage open Web-based crowdsourcing platforms (e.g., Amazon Mechanical Turk, Prolific). These platforms provide a large pool of freelance crowd workers who are available 24/7 and can complete assigned tasks at affordable costs [28]. However, the response time of crowd workers may vary due to potential response delays and task dropouts [52]. Therefore, it is essential to design a human-AI co-learning system to improve the label quality while minimizing the crowd response delay.

In order to address the aforementioned challenges, this paper presents *SymLearn*, a symbiotic human-AI co-learning framework that explores the complex interdependence between AI and HI to harness their collective strengths in solving the WebHA problem. In particular, we first design a novel search space for crowd-AI collaboration, which provides flexible options for SymLearn to identify a collective set of AI models and crowd workers that are capable of jointly estimating the WebHA label for each input image. Subsequently, we develop a context-aware multi-armed bandit model that selects an optimized set of WebHA models and crowd workers from the search space to accurately determine the WebHA label for each input image while minimizing the delay in crowd responses. To the best of our knowledge, SymLearn is the first symbiotic human-AI co-learning framework that develops a mutually beneficial human-AI collaboration design to effectively solve the WebHA problem. It is worth noting that SymLearn has the potential to address a broader range of Web-based applications beyond WebHA (e.g., fake news detection, abnormal event identification, sentiment analysis, and opinion mining) by jointly exploring the imperfect yet complementary AI and human intelligence. We demonstrate the effectiveness of SymLearn by evaluating it through two real-world WebHA applications, namely, mask wearing policy adherence and social distancing policy adherence. The results show that SymLearn outperforms the state-of-the-art deep learning approaches, human-AI models, and AI model optimization frameworks by improving WebHA performance and reducing crowd response delay.

## 2 RELATED WORK

**Social Media for Common Goods.** Web-based social media have attracted widespread interest in recent years due to their unparalleled capacity for obtaining real-time situational awareness in many real-world applications for common goods [2, 4, 17, 25, 42, 47, 54]. For example, Trinh *et al.* leveraged a multi-access mobile social network and a sequential deep neural network-based prediction model to detect urban anomaly with high reliability and low latency [42]. Hao *et al.* designed a multimodal neural network-based disaster damage assessment framework that utilizes massive social media text and image data to classify disaster damage types in hurricane events [17]. Alomari *et al.* utilized an automatic labeling method to develop a distributed machine learning-based model for traffic-related event detection from social media data [2]. However, it remains a critical challenge to integrate AI and human intelligence to solve the WebHA problem that leverages social media for public health, all while considering the complex interdependence between AI and HI. In this paper, we develop a novel symbiotic human-AI

co-learning system to accurately assess public healthcare policy adherence by utilizing massive social sensing imagery data.

**Public Healthcare Assessment.** With the emergence of diseases like COVID-19, Polio, and Monkeypox, the public health and well-being of society are becoming increasingly threatened, particularly among vulnerable groups such as the elderly and those with weakened immune systems [5]. Public healthcare assessment is a crucial aspect of healthcare management and policy-making that has garnered a significant amount of attention [1, 8, 23, 31, 35, 49]. For example, Raza *et al.* created a voice-based social media platform to provide trustworthy health information during COVID-19 to underserved online communities [35]. Chen *et al.* developed a deep neural network model to predict health risks accurately by utilizing socioeconomic status and environmental factors from social media [8]. Yue *et al.* proposed a contrastive domain adaptation approach for early detection of misleading healthcare information on social media [49]. This paper studies a novel problem of utilizing social media imagery data for accurate healthcare policy adherence assessment (i.e., WebHA). In particular, we develop a novel symbiotic human-AI co-learning framework to improve the overall performance of WebHA applications.

**Human-AI Co-Learning.** Our SymLearn is also related to recent progress in developing human-AI co-learning solutions to combine the complementary strengths of AI and HI [21, 38, 48, 52]. For example, Sener *et. al.* developed a deep core-set approach that selects a subset of representative images to collect crowd labels for AI model retraining to improve the overall image classification performance [38]. Zhang *et al.* utilized crowd labels from non-expert citizen scientists with deep damage assessment models to enhance the performance of disaster damage severity classification in disaster response [52]. Yoo *et al.* proposed a parametric crowd-AI hybrid solution that employed a task-agnostic prediction loss design to improve the AI and crowd intelligence fusion for accurate image classification [48]. Hu *et al.* built a crowdsourcing-based image data bias detection framework that improves the image classification performance of deep visual models by identifying sampling bias [21]. However, current human-AI co-learning solutions focus on leveraging one type of intelligence to improve the other while overlooking the interdependence and interactions between them. In contrast, our SymLearn develops a closed-loop and mutually beneficial human-AI collaboration framework to jointly optimize the WebHA performance.

## 3 PROBLEM FORMULATION

To define the human-AI collaborative WebHA problem, we first introduce several basic concepts.

DEFINITION 1. **Social Media Image ($X$)**: We define $X$ as a collection of images related to public health crises that are gathered from social media to evaluate the adherence to healthcare policies. Here, $X_i$ denotes the $i^{th}$ image within the set, and $I$ represents the total number of images involved in the studied WebHA application.

DEFINITION 2. **WebHA Label ($Y$)**: In this paper, we focus on a classification-based WebHA application, where the public health policy adherence status is classified into $B$ different categories. For example, in a WebHA application of determining the mask wearing compliance from CDC [6], the WebHA labels are binary: *adhering*

and *not adhering*. Specifically, we define $Y = \{Y_1, Y_2, ..., Y_I\}$ as the ground-truth WebHA labels for all studied WebHA images, where $Y_i$ represents the WebHA label assigned to $X_i$.

DEFINITION 3. **WebHA Model ($M$)**: we define $M = \{M_1, M_2, ..., M_K\}$ to be a set of AI models (e.g., CNN, Transformer) for WebHA tasks. In particular, we denote $\widehat{Y_i^{M_k}}$ as the WebHA label estimated by a WebHA model $M_k$ for the social media image $X_i$.

We note that AI-based WebHA models can generate incorrect WebHA labels for input images and it is sometimes challenging to troubleshoot AI models without human interventions [37]. Therefore, our SymLearn integrates human intelligence from crowdsourcing systems with AI models in a symbiotic co-learning system to improve WebHA estimation accuracy. In particular, the practice of wearing masks and adhering to social distancing guidelines has gained widespread acceptance and is embraced by a majority of the population during the COVID-19 pandemic. Consequently, individuals have become more proficient at assessing compliance with mask wearing and social distancing policies [15]. We further define a few terms on the crowdsourcing aspect of our model below.

DEFINITION 4. **Crowd Worker ($C$)**: Our SymLearn framework recruits freelance workers from crowdsourcing platforms to estimate the WebHA label for an input image, which will be integrated with output from the WebHA model to generate the final WebHA label for the image. Each crowd worker is assigned the task of providing a label for each studied image. Specifically, for an image $X_i$, a group of $N_i$ crowd workers denoted by $C_i = \{C_1^i, C_2^i, \ldots, C_{N_i}^i\}$ are assigned by SymLearn to label $X_i$. We define $\widehat{Y_i^{C_n^i}}$ as the corresponding WebHA label from crowd worker $C_n^i$ for image $X_i$.

DEFINITION 5. **Crowd Response Time ($D$)**: We define $D_i$ as the duration between the start time of the first crowd worker and the completion time of the last crowd worker for the $i^{th}$ image to complete the labeling task for the social media image $X_i$.

DEFINITION 6. **WebHA Label Collectively Learned by Crowd and AI ($\widehat{Y}$)**: $\widehat{Y}$ refers to the final outputs of SymLearn, which are generated by our symbiotic human-AI system that integrates inputs from WebHA models and crowd workers to improve overall label quality and reduce response delay. We will discuss the detailed design of our SymLearn in the next section. In particular, $\widehat{Y_i}$ denotes the final WebHA label identified for the social media image $X_i$.

In our human-AI collaborative WebHA problem, the objective is to dynamically identify a set of WebHA models and crowd workers that can collaboratively learn the accurate WebHA label for each input image while minimizing the response time of the crowd worker. We formally define our problem as follows:

$$\underset{\widehat{Y_i}}{\arg\max} \left( \Pr(\widehat{Y_i} = Y_i \mid X_i, M, C_i) \right), \forall\, 1 < i < I$$
$$while\ \underset{\widehat{Y_i}}{\arg\min} \left( D_i \mid X_i, M, C_i \right), \forall\, 1 < i < I \qquad (1)$$

Our problem is challenging due to the complex interdependence between HI and AI and the intricate trade-off between label quality and response delay in the human-AI co-learning framework.

# 4 SOLUTION

In this section, we present our *SymLearn*, a symbiotic crowd-AI co-learning framework to address the healthcare policy adherence assessment problem. We provide an overview figure of SymLearn in Appendix A.1. In particular, SymLearn contains two key modules:

(1) *Symbiotic Crowd-AI Collaboration Space Design (SCSD)*: it introduces a crowd-AI collaboration search space design that offers adaptable options for SymLearn to systematically investigate optimal combinations of diverse AI models and crowd workers for the joint estimation of WebHA labels for each input image. In particular, our module includes a novel crowdsourcing interface design where the AI model can guide humans to pay attention to fine-grained details crucial for determining WebHA labels, thereby improving the accuracy of human label quality.

(2) *Active Integrated Crowd-AI Co-Learning (AICC)*: it develops a novel context-aware multi-armed bandit model that selects a set of optimized WebHA models and crowd workers from the search space in the SCSD module. The AICC then integrates the inputs from the selected WebHA models and crowd workers to identify the accurate WebHA label for each input image while minimizing the crowd response delay.

## 4.1 Symbiotic Crowd-AI Collaboration Space Design

In the first subsection, we present our symbiotic collaboration space design by exploring the collective power of both crowdsourcing and AI. We first define a key definition in our SCSD module.

DEFINITION 7. **Crowd-AI Collaboration Strategy**: We define a crowd-AI collaboration strategy as a scenario where SymLearn selects a set of WebHA models and a certain number of crowd workers with a group of assigned crowdsourcing questions to estimate the WebHA label for an input image. Our SymLearn aggregates the inputs from both WebHA models and crowd workers to generate the final WebHA label for each image.

The goal of our SymLearn is to identify the desirable crowd-AI collaboration strategy that produces correct WebHA labels for each input image. In particular, our crowd-AI collaboration strategy contains three "control knobs" in both crowd and AI space to jointly optimize the performance of WebHA tasks: i) selecting a different set of WebHA models for each input image; ii) selecting different number of crowd workers for each input image; iii) determining the crowdsourcing questions sent to each crowd worker. Such a design provides an effective exploration of the combination of WebHA models, crowd workers, and appropriate crowdsourcing questions to collectively generate accurate WebHA labels while minimizing crowd response delays. An illustration of our crowd-AI collaboration strategy design is shown in Figure 2. Note that our SCSD module focuses on designing a crowd-AI collaboration space that includes all possible options for SymLearn to establish a crowd-AI collaboration strategy for each input image. We will present a novel crowd-AI co-learning framework in the next subsection to explore the trade-off between the above control knobs and determine the optimal crowd-AI collaboration strategy for each input image.

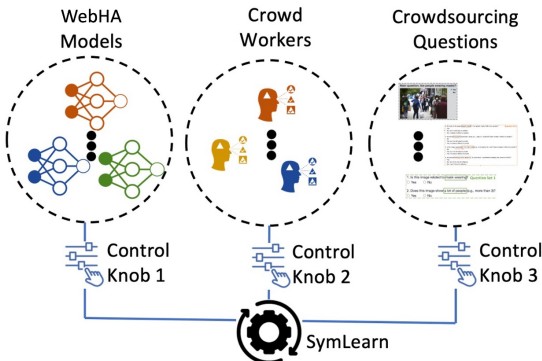

**Figure 2: Illustrations of Crowd-AI Collaboration Strategy**

First, our crowd-AI collaboration strategy provides the option to select different WebHA models for each input image. In particular, we consider a WebHA model that contains two key attributes: *network architecture* and *hyperparameter configuration*, where the selection of the two attributes will directly affect its WebHA performance. More specifically, in terms of network architecture, we consider the WebHA models with different network architectures in terms of the type of the convolutional block (residual block or dense block), the number of convolutional layers per block, the width of convolutional block, the growth rate, and the size of input features [50]. In terms of hyperparameter configurations, we consider the WebHA models with different learning rates, optimizers (e.g., RMSprop, SGD, Adam), weight decays, and conditional parameters of the optimizer (e.g., RMSprop alpha, SGD momentum, Adam beta1, Adam beta2) [27]. Second, our crowd-AI collaboration strategy provides the option to select different numbers of crowd workers for each input image. For example, a higher level of inconsistency between the outputs of WebHA models often indicates the need of a larger number of crowd workers. However, there also exists a non-trivial trade-off between the number of crowd workers and the crowdsourcing budget as well as the potential delay in crowd responses. We will discuss our context-aware multi-armed bandit model to address such a challenge next.

Finally, our crowd-AI collaboration strategy provides the option to ask different levels of crowdsourcing questions for a given input image to assist crowd workers in obtaining the correct labels as illustrated in Figure 3. Specifically, our crowdsourcing question design starts with a basic question group 1 (i.e., questions 1 and 2) to ensure that crowd workers pay close attention to the context in the images when they provide labels. Subsequently, we design question group 2 (i.e., questions 3-6) to encourage workers to consider the crucial factors related to WebHA prediction tasks. For instance, question 4 prompts workers to consider the impact of occlusion on the prediction accuracy. Finally, we include question group 3 (i.e., question 7) to prompt workers to re-evaluate their predictions and assess whether they would like to change their predictions based on the previous questions they answered. The design of our crowdsourcing questions is motivated by the concept of "self-reflection" from psychology, which has been demonstrated to be an effective metacognitive mechanism in improving one's performance in cognitive tasks [9]. We refer to our designed crowdsourcing questions

**Question Group 1**

1. Is this image related to mask-wearing?
○ Yes
○ No

2. Does this image show a lot of people (e.g., more than 3)?
○ Yes
○ No

**Main Question**

Main question: Are people wearing masks?
◉ Yes
○ No

**Question Group 2**

3. Do most of the people face forward? If people don't face forward, does this make it difficult for you to predict?
○ Yes
○ No, but it is still easy to predict
○ No, it makes it difficult to predict

4. Are there occlusions of people's faces (e.g., other people or objects)? If there are occlusions, does it make it difficult for you to predict?
○ No
○ Yes, but it is still easy to predict
○ Yes, it makes it difficult to predict

5. Is the image captured far from the crowd (e.g., faces look small or blurred)? Do you think it makes it difficult for you to predict?
○ No
○ Yes, but it is still easy to predict
○ Yes, it makes it difficult to predict

6. Are people wearing masks properly (e.g., fully covering mouths and noses)? Do you think it is an effective wearing of masks that prevents the COVID?
○ Yes
○ No, but I think it is still effective
○ No, I don't think it is effective

**Question Group 3**

7. Based on your answers to questions 1 - 6, would you like to change your prediction of mask-wearing in the main question?
○ Yes
○ No

**Figure 3: An Example of AI-assisted Crowdsourcing Task in SCSD Module**

(i.e., question groups 1-3) as *"self-reflection" questions*. Unlike current crowdsourcing approaches that rely on passively receiving raw annotations from crowd workers (e.g., crowd labels of simple annotation tasks without any active quality control) [13], our design actively guides the crowd workers to reflect on their labels during their reasoning process. The "self-reflection" questions serve as options for SymLearn to stimulate the crowd workers to think more carefully and provide high-quality labels when necessary for the images that are proven to be challenging to the WebHA models. We also evaluate the effectiveness of our AI-assisted crowdsourcing task design and demonstrate the advantage of our design in improving the crowd label quality in Subsection 5.3.2.

## 4.2 Active Integrated Crowd-AI Co-Learning

In our second module, we introduce an active integrated crowd-AI co-learning (AICC) module that identifies the desirable collaboration strategy from the search space defined in the SCSD module. In AICC, WebHA models and crowd workers work collaboratively to learn the accurate WebHA label of a given social media image while minimizing the response delay of the crowd. To achieve such an objective, we develop a novel context-aware multi-armed bandit (CMAB) model to determine the desirable crowd-AI collaboration strategy to identify the correct label for each input image. We refer to our model as *Crowd-AI CMAB*.

The multi-armed bandit model is a classic problem in decision-making that involves choosing between multiple options (the "arms" of the bandit) with uncertain reward probabilities in order to maximize the cumulative reward over time [24]. It balances the exploration of less-known options with the exploitation of those that have shown high rewards so far. The context-aware multi-armed bandit model is a variation of the classic multi-armed bandit model by taking into account contextual information when making decisions. In a context-aware MAB setting, an agent interacts with arms to maximize cumulative rewards over time. By incorporating contextual information into MAB, the agent is better equipped to make informed decisions that adapt to the changing environment, leading to improved performance and a more efficient exploration-exploitation trade-off. We first introduce a few key definitions for the Crowd-AI CMAB model.

DEFINITION 8. **Arms ($A$)**: We define the set of arms, denoted as $A = \{A_1, A_2, ..., A_J\}$, within our Crowd-AI CMAB model representing different collaboration strategies between crowd workers and AI models for WebHA tasks. Specifically, $A_j$ represents the $j^{th}$ arm, and $J$ represents the total count of available arms. Our model's objective is to determine the desirable collaboration strategy for each input image.

DEFINITION 9. **Context ($O$)**: Our definition of the context $O$ is the relevant information needed to make the desirable decision regarding collaboration between crowd and AI. In particular, the context includes the entropy of AI model predictions for an input image. The entropy of predictions generated by various AI models represents the inconsistency between the AI outputs and serves as a measure of the difficulty of the prediction task: when AI models are less consistent with each other, it indicates the task to be

more challenging, as there are multiple possible predictions with high probabilities. Intuitively, incorporating the entropy into the context helps us identify difficult cases for AI models, recruit more crowd workers, and assign navigation questions to improve overall prediction performance.

DEFINITION 10. **Reward ($R$)**: The reward $R$ is a feedback mechanism that our SymLearn receives by choosing a specific crowd-AI collaboration strategy (arm). In our paper, we aim to maximize the prediction accuracy of WebHA while minimizing the response delay. To achieve this goal, we incorporate two components in the definition of the reward. The first component is the correctness $Q_j^i$ of the WebHA label estimated by our SymLearn framework (definition 6). The second component is the response delay $D_j^i$ that refers to the duration between the start time of the first crowd worker and the completion time of the last crowd worker for the $i^{th}$ image and $j^{th}$ arm. Specifically, the reward $R_j^i$ for the $i^{th}$ image and $j^{th}$ arm is formally defined as a combination of the above two components:

$$R_j^i = e^{Q_j^i} + e^{-D_j^i} \tag{2}$$

The above reward function design ensures that our SymLearn is rewarded when it selects the collaboration strategy that maximizes the generated WebHA label quality for the training data while minimizing the crowd response delay as we discuss in the overall objective of our problem (Equation (1)).

The objective of the CMAB model is to maximize cumulative reward for all studied images. To achieve this goal, we begin by modeling the reward distribution as a linear function of the context vector [53]. Using this approach, we can calculate the expected reward for selecting the $j^{th}$ arm for the $i^{th}$ image as follows:

$$\widehat{R_j^i} = \beta_j O_i \tag{3}$$

where $\beta_j$ represents the arm-specific parameter vector, and $O_i$ is the context information for the $i^{th}$ image.

The key challenge of estimating unknown parameter $\beta_j$ to learn the reward $\widehat{R_j^i}$ is to strike a good balance between exploration (i.e., trying new crowd-AI collaboration strategies) and exploitation (i.e., selecting crowd-AI collaboration strategies with high expected rewards). While exploring new strategies of the crowd and AI collaboration obtains more information on how different collaboration strategies contribute to the reward, it also requires an additional crowdsourcing budget and leads to increased response delay. Conversely, exploitation limits the exploration of new collaboration strategies to meet the constrained crowdsourcing budget, but it may miss the collaboration strategies with high rewards. Therefore, it remains to be a challenging question on how to achieve the optimal balance between exploration and exploitation of the collaboration strategies between crowd and AI.

To address the above challenge, our AICC module incorporates an upper confidence bound (UCB) estimation, which selects the arm with the highest upper confidence bound based on the estimated reward distribution and confidence intervals [16]. Specifically, our AICC module utilizes a tree structure, in which each node represents a distinct set of crowd-AI collaboration strategies with a specific context as described in Definition 9. Our AICC module selects a node in the tree structure based on the current context and then estimates the UCB of the arms within that node. More specifically,

our AICC module traverses the tree to identify the node $v$ that corresponds to the current context $O_i$. The node $v$ corresponds to a partition of the arms $A_v$, which are the set of arms that have similar expected rewards for the given context features. Our AICC module then applies the UCB estimation to the arms in $A_v$, using the UCB value for the $i^{th}$ image and $j^{th}$ arm given by:

$$UCB_j^i = \widehat{R_j^i} + \sqrt{\frac{\alpha \log(\sum_{q \in A_v} n_q^i)}{n_j^i}} \tag{4}$$

where $\widehat{R_j^i}$ is the estimated reward for image $i$ and arm $j$, $n_j^i$ is the number of times arm $j$ has been selected for image $i$, and $\alpha$ is a tuning parameter that controls the degree of exploration.

After the aforementioned process, we select the crowd-AI collaboration strategy that yields the highest reward for each input image and aggregates the WebHA labels generated by both the crowd and AI to produce the final outputs of our SymLearn framework. As the WebHA models and crowd workers vary for each input image, advanced label aggregation strategies (e.g., estimation-theoretic approaches, active learning) [20], are not applicable to our problem as they often rely on a fixed set of aggregation sources (e.g., WebHA models and crowd workers) to establish the aggregation model. Therefore, we adopt majority voting as our simple but robust aggregation function. In addition, we provide a summary of SymLearn using pseudocode in Appendix A.2.

## 5 EVALUATION

### 5.1 Datasets and Crowdsourcing Settings

*5.1.1 Two Real-world WebHA Applications.* We conduct experiments to evaluate the performance of SymLearn using two publicly available WebHA datasets published by [29]: 1) Mask Wearing Policy Adherence (MWPA) application and 2) Social Distancing Policy Adherence (SDPA) application. Both datasets comprise a collection of images related to public health crisis, sourced from a popular social media platform (i.e., Twitter/X [45]) during the COVID pandemic. In the MWPA dataset, the WebHA labels are classified into two categories: adhering (i.e., all individuals in the picture are appropriately wearing face masks in line with mask wearing guidelines from CDC [6]) and not adhering (i.e., not all individuals in the image are wearing face masks correctly). In the SDPA dataset, the WebHA labels are also classified into two categories: adhering (i.e., all individuals in the picture are following social distancing guidelines according to CDC's recommendations [7]) and not adhering (i.e., not all individuals in the image are practicing social distancing). In our experiments, we use the ground-truth labels provided by [29] for the evaluation. To ensure the accuracy of the ground truth labels and the reliability of our evaluation, we further enlist the help of domain experts to only include the images where the ground-truth labels of mask wearing policy adherence and social distancing policy adherence can be clearly validated by the domain experts. Table 1 provides the statistics of the two datasets. In addition, we split each dataset into training, validation, and testing sets, following a ratio of 6:2:2, as outlined in [32]. In particular, SymLearn and compared baselines are trained on the training and validation sets for WebHA tasks. The testing set is then used to

evaluate the performance of SymLearn and compared baselines for WebHA applications.[1]

**Table 1: Statistics of Two WebHA Applications**

| Application | MWPA | SDPA |
|---|---|---|
| Data Collection Time | May, 2020 | May, 2020 |
| Number of Images | 2,165 | 1,027 |
| % of Class *Adhering* | 75.4% | 41.0% |
| % of Class *Not Adhering* | 24.6% | 59.0% |

*5.1.2 Crowdsourcing Settings.* To recruit crowd workers for our experiments, we employ Amazon Mechanical Turk (AMT) [44], a commonly used crowdsourcing platform. To ensure high-quality annotations, we impose two requirements for selecting participants in our task: a minimum of 1,000 approved tasks completed by the worker and an overall approval rate above 95%. In the experiments, 479 crowd workers are recruited for the MWPA application and 232 workers are recruited for the SDPA application. The average WebHA annotation accuracy for the crowd workers recruited for the MWPA and SDPA applications are 92.8% and 80.9%, respectively. We compensate each crowd worker $0.05 per annotation task for an image. We followed the IRB protocol of this project.

## 5.2 Baselines and Experimental Settings

To evaluate the proposed SymLearn, we compare it with a rich collection of state-of-the-art baselines, including: (1) *Deep Neural Network (DNN):* **ResNet** [19], **DenseNet** [22], and **DeiT** [41]; (2) *Crowd-AI Collaboration:* **Deep Active** [38], **CrowdLearn** [52], and **LL++** [39]; (3) *AI Model Optimization:* **DEHB** [3], **BOHB** [10], and **MnasNet** [40]. A detailed description of all compared baselines is presented in Appendix A.3.

To ensure a fair comparison, we use the same input to all compared baselines: 1) the images collected from the WebHA applications; 2) the ground-truth labels for the WebHA images in the training and validation sets; 3) the labeled WebHA images from the crowdsourcing query. Specifically, we use the queried crowd labels to fine-tune the DNN and AI model optimization baselines so that all compared schemes have the same inputs and the performance of compared baselines is optimized. We also include the detailed hyperparameter settings for our experiment in Appendix A.4.

We utilize four commonly used evaluation metrics for imbalanced image classification tasks in our experiments. The metrics include Accuracy (Acc.), F1-Score, Kappa Score ($\mathcal{K}$-Score), and Matthews Correlation Coefficient (MCC). In particular, we include $\mathcal{K}$-Score and MCC in addition to Accuracy and F1-Score, since the MWPA and SDPA datasets used in our study are imbalanced, and $\mathcal{K}$-Score and MCC are shown to be reliable metrics for evaluating imbalanced classification tasks. A higher value for these metrics indicates a better classification result for WebHA tasks.

## 5.3 Evaluation Results

*5.3.1 WebHA Classification Performance.* We first evaluate the classification performance of all compared approaches on the MWPA and SDPA applications. The evaluation results are summarized in

---

[1]Our codes will be made publicly available upon the acceptance of the paper.

Table 2 and 3. We observe that SymLearn clearly outperforms all compared schemes on all evaluation metrics. For example, on the MWPA dataset, the performance gains of SymLearn compared to the best-performing baseline (i.e., BOHB) on Accuracy, F1-Score, K-Score, and MCC are 6.93%, 5.29%, 13.54%, and 12.49%, respectively. Such performance gains are primarily achieved by our symbiotic crowd-AI collaborative learning framework that effectively identifies a desirable crowd-AI collaboration strategy to estimate the accurate WebHA label for each input image through the novel context-aware multi-armed bandit model. The consistent performance gains achieved by SymLearn over two diversified WebHA applications (i.e., MWPA and SDPA) demonstrate the robustness of SymLearn over different WebHA applications.

**Table 2: Performance Comparisons on MWPA**

| Algorithm | Acc. | F1 | $\mathcal{K}$-Score | MCC |
|---|---|---|---|---|
| ResNet | 0.7968 | 0.8629 | 0.4704 | 0.4713 |
| DenseNet | 0.7852 | 0.8526 | 0.4585 | 0.4619 |
| DeiT | 0.8106 | 0.8758 | 0.4779 | 0.4786 |
| Deep Active | 0.8314 | 0.8962 | 0.4606 | 0.4972 |
| CrowdLearn | 0.8453 | 0.8942 | 0.6076 | 0.6114 |
| LL++ | 0.8291 | 0.8840 | 0.5600 | 0.5620 |
| DEHB | 0.8522 | 0.8984 | 0.6284 | 0.6336 |
| BOHB | 0.8545 | 0.8976 | 0.6497 | 0.6639 |
| MnasNet | 0.7898 | 0.8530 | 0.4882 | 0.4968 |
| **SymLearn** | **0.9238** | **0.9505** | **0.7851** | **0.7888** |

**Table 3: Performance Comparisons on SDPA**

| Algorithm | Acc. | F1 | $\mathcal{K}$-Score | MCC |
|---|---|---|---|---|
| ResNet | 0.7122 | 0.6911 | 0.4289 | 0.4400 |
| DenseNet | 0.7073 | 0.6809 | 0.4162 | 0.4243 |
| DeiT | 0.6829 | 0.6829 | 0.3859 | 0.4118 |
| Deep Active | 0.7121 | 0.6629 | 0.4125 | 0.4135 |
| CrowdLearn | 0.7024 | 0.6806 | 0.4096 | 0.4201 |
| LL++ | 0.7317 | 0.7150 | 0.4695 | 0.4838 |
| DEHB | 0.6927 | 0.6957 | 0.4068 | 0.4373 |
| BOHB | 0.7463 | 0.7045 | 0.4831 | 0.4847 |
| MnasNet | 0.6683 | 0.6495 | 0.3452 | 0.3566 |
| **SymLearn** | **0.7951** | **0.7439** | **0.5733** | **0.5738** |

*5.3.2 Crowd Response Quality Comparison.* In this set of experiments, we evaluate the effectiveness of our AI-assisted crowdsourcing task design as shown in Figure 3. In particular, we compared the crowd label quality on WebHA returned by the crowd workers with and without guidance from the "Self-Reflection" Questions

(i.e., Question Group 1-3 as shown in Figure 3) in both MWPA and SDPA applications. The evaluation results are presented in Figure 4. We observe that the quality of crowd labels has clearly improved with the self-reflection question design (i.e., "w/SRQ" in Figure 4) compared to those without the self-reflection question design (i.e., "w/o SRQ" in Figure 4). In our experiments, we recruit the same amount of crowd workers to perform labeling tasks on the same set of studied images for both the "w/SRQ" and "wo/SRQ" settings. The evaluation results demonstrate the effectiveness of our AI-assisted crowdsourcing task design that stimulates crowd workers to reflect on their labels during their reasoning process and provide high-quality labels when necessary for images that are proven to be challenging to the WebHA models. In general, the first two sets of evaluation results above demonstrate that SymLearn effectively harnesses the symbiotic relationship between AI and HI for WebHA tasks: Subsection 5.3.1 demonstrates that SymLearn leverages crowd inputs to optimize AI model performance, while Subsection 5.3.2 shows that SymLearn utilizes AI to guide crowd workers in enhancing crowd label quality.

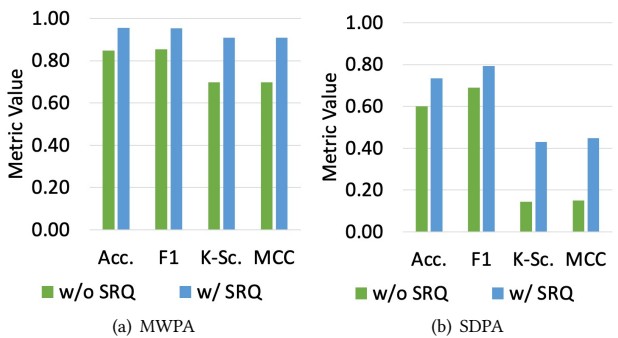

(a) MWPA                    (b) SDPA

**Figure 4: Crowd Response Quality Comparison (SRQ: "Self-Reflection" Questions)**

*5.3.3 Crowd Response Time.* We then evaluate the crowd response time (Definition 5) of SymLearn and the crowd-AI collaboration baselines (i.e., Deep Active, CrowdLearn, LL++). The evaluation results are presented in Figure 5. We note that the response time of SymLearn framework is the least for both MWPA and SDPA tasks compared to other baselines. Such a performance gain is due to the novel context-aware multi-armed bandit model design that identifies the smallest number of crowd labels needed to integrate with the AI outputs to generate the accurate WebHA label for each input image. The improvement in crowd response time clearly expedites the delivery of WebHA results in a timely manner, which allows public agencies to receive timely information and effectively implement and adjust policies.

*5.3.4 Ablation Study.* Finally, we conduct an ablation study to assess the contributions of two key modules, namely SCSD and AICC, in our SymLearn framework. We present the performance evaluation results obtained by eliminating each of these modules individually. In particular, we replace the SCSD module by directly tasking the crowd workers to provide the WebHA label without our crowd-AI collaboration space design as shown in Figure 3. In

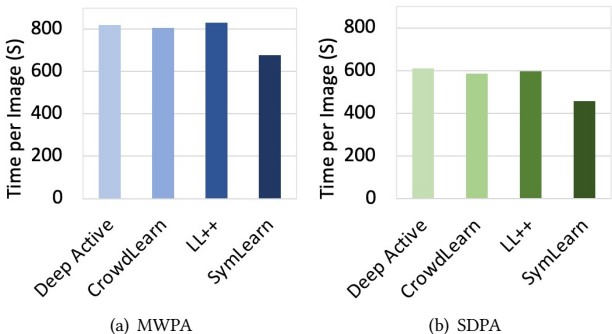

(a) MWPA                    (b) SDPA

**Figure 5: Crowd Response Time**

addition, we remove AICC model by integrating the inputs from AI models and crowd workers through a consensus-based label aggregation [20]. The evaluation results, presented in Figure 6, show a decrease in performance in terms of all evaluation metrics after removing SCSD or AICC modules on both datasets. The results clearly demonstrate that both the SCSD and AICC modules make critical contributions to the SymLearn framework in terms of WebHA prediction accuracy.

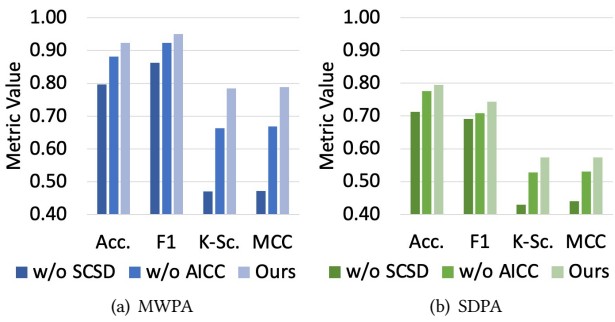

(a) MWPA                    (b) SDPA

**Figure 6: Ablation Study**

## 6 CONCLUSION

This paper introduces a SymLearn framework to solve the WebHA problem. In particular, our SymLearn develops a symbiotic human-AI co-learning framework that explores the complex interdependence between AI and HI to harness their collective strengths in addressing the WebHA problem. Our SymLearn system addresses the limitation of current human-AI systems which often focus on leveraging one type of intelligence to improve the other while overlooking the interdependence and interactions between them. Our SymLearn is shown to achieve the highest WebHA accuracy compared to a broad set of baselines in two real-world WebHA applications while minimizing the crowd response delay. We believe SymLearn provides useful insights to develop a closed-loop and mutually beneficial human-AI collaboration framework in addressing a broader range of real-world Web-based applications beyond WebHA, such as fake news detection, abnormal event identification, sentiment analysis, and opinion mining.

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

# A APPENDIX

## A.1 Overview of SymLearn

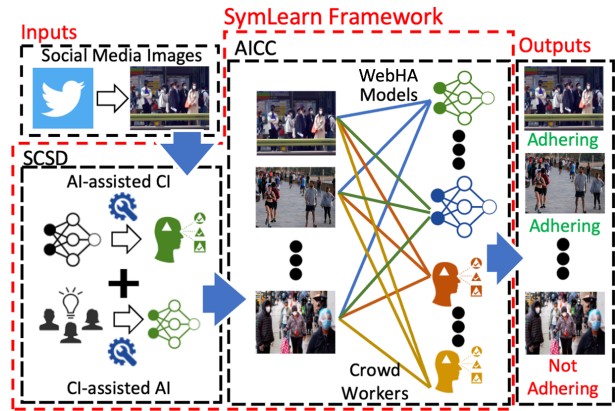

**Figure 7: Overview of SymLearn Framework**

## A.2 Summary of SymLearn Framework

We summarize the SymLearn framework in Algorithm 1. The inputs to the framework are the social media images $X$, crowd workers $C$, and WebHA models $M$. The outputs to the framework are the generated crowd-AI collaborative label $\widehat{Y}$ for WebHA applications with the desirable collaboration strategy between crowd workers and WebHA models.

---

**Algorithm 1** SymLearn Framework Summary

---

1: **input**: $X$, $C$, $M$
2: **output**: $\widehat{Y}$
   ▷ *Training Phase*
3: Calculate context $O_i$ for each $X_i$ (Definition 9)
4: **for** each image $X_i$ **do**
5:   **for** each arm $A_j$ **do**
6:     Estimate reward $R_j^i$ (Definition 10)
7:   **end for**
8: **end for**
   ▷ *Prediction Phase*
9: **for** each image $X_i$ **do**
10:   Predict arm $A_j$ (Definition 8)
11:   Add crowd workers and questions based on $A_j$
12:   Collect crowd prediction $\widehat{Y_i^{C_i}}$ (Definition 4)
13:   Use $\widehat{Y_i^M}$ and $\widehat{Y_i^{C_i}}$ to generate collaborative prediction $\widehat{Y_i}$ and add it to $\widehat{Y}$ (Definition 6)
14: **end for**

---

## A.3 Baselines

To evaluate the proposed SymLearn, we compare it with a rich collection of state-of-the-art baselines as follows:

1) *Deep Neural Network (DNN):*
- **ResNet** [19]: a deep convolutional neural network architecture that introduces residual block with skip connections to help improve image classification accuracy.

- **DenseNet** [22]: a densely connected convolutional neural network that connects each layer to every other layer to promote feature learning for image classification.
- **DeiT** [41]: a vision transformer that leverages self-attention mechanisms to achieve state-of-the-art image classification performance with fewer parameters than traditional convolutional neural networks.

2) *Crowd-AI Collaboration:*
- **Deep Active** [38]: an active learning technique that identifies a subset of data samples for crowd labeling and subsequently incorporates the crowd labels to retrain the AI model to enhance WebHA classification performance.
- **CrowdLearn** [52]: a crowd-AI collaborative framework that utilizes the power of crowdsourced human intelligence to troubleshoot AI models and enhance the overall classification performance.
- **LL++** [39]: a crowd-AI hybrid approach that leverages a crowdsourcing-based uncertainty-aware estimation model to identify and resolve failure cases of AI models in image classification.

3) *AI Model Optimization:*
- **DEHB** [3]: a representative optimizer for AI models that designs a strategy based on non-stochastic infinite-armed bandit to optimize AI model performance.
- **BOHB** [10]: a commonly used optimizer for AI models that utilizes a Bayesian optimization approach to enhance AI model performance.
- **MnasNet** [40]: a lightweight technique to optimize AI models by leveraging a multi-objective reinforcement learning scheme for factorized hierarchical AI model design search.

## A.4 Hyperparameter Settings

In the experiments, our SymLearn framework and compared baselines are implemented using PyTorch 1.1.0 libraries [34] and trained on NVIDIA Quadro RTX 6000 GPUs. For DNN and crowd-AI baselines, we optimize the parameters of each compared scheme on the training and validation datasets to achieve their best performance [18]. For AI model optimization baselines, we follow the standard practice in AI model optimization to optimize the AI model design using the training and validation datasets [11].

In our experiments, we follow a standard AI model optimization process [26, 50] to define the network architecture and hyperparameter search space. Specifically, the network architecture search space includes 1) the types of convolutional block (residual block or dense block), 2) the number of convolutional layers per block (ranging from 1 to 36), 3) the width of convolutional block (between 21 and 27), 4) the growth rate (between 32 and 48), and 5) the size of input features (between 64 and 96). For hyperparameter configurations, we consider the search space that includes 1) the learning rate (between $10^{-6}$ and $10^{-3}$), 2) the weight decay (between 0 and $10^{-3}$), 3) three candidate optimizers (SGD, RMSprop, and ADAM), 4) the conditional parameters of SGD momentum, RMSprop alpha, Adam beta1, and Adam beta2 (between 0.8 and 1.0), and 5) the number of epochs (between 30 and 150).

