# OpenReview forum: "SymLearn: A Symbiotic Crowd-AI Collective Learning Framework to Web-based Healthcare Policy Adherence Assessment"
_ACM.org/TheWebConf/2024/Conference — TheWebConf24_

### Official Review · Reviewer_GbjJ · 2023-11-19

**Novelty:** 4
**Technical Quality:** 4

**Review:**

The paper introduces SymLearn, a novel framework that integrates human and AI intelligence for assessing healthcare policy adherence using social media imagery. It proposes a symbiotic approach that leverages the strengths of both crowdsourcing and AI models to address the Web-based Healthcare Policy Adherence (WebHA) problem.


Quality: The paper stands out in its innovative approach to combining human and AI intelligence, addressing the complex problem of WebHA assessment. The methodology is comprehensive, integrating various aspects of crowdsourcing, AI modeling, and decision-making processes.

Clarity: The paper is well-structured, providing clear explanations of the framework, methodologies, and experimental setup. The use of illustrative examples and figures enhances understanding.


Originality: The concept of a symbiotic human-AI collective learning framework is original and represents a novel approach to the problem. The integration of crowdsourcing and AI in this context is an innovative step forward.

Significance: The research has significant implications for healthcare policy adherence assessment, particularly in leveraging social media data. It offers valuable insights for policy-makers, healthcare professionals, and AI researchers.


Cons:

There seem to be potential limitations in the scalability and generalizability of the SymLearn framework across various social media platforms and different health crisis contexts. Can you discuss how the framework might be adapted to diverse platforms and crisis situations?

The experimental comparisons and evaluation metrics employed in the study seem to lack comprehensiveness, particularly in the context of potentially imbalanced datasets. Could you provide a rationale for the chosen metrics and discuss the possibility of incorporating more diverse comparison methods in future studies? Specifically, including metrics like AUROC (Area Under the Receiver Operating Characteristic curve) and AUPRC (Area Under the Precision-Recall Curve) could offer more insights, especially for analyzing imbalanced datasets. How might these metrics enhance the evaluation of your framework?

I noticed that the current set of comparative analyses in the study might benefit from a broader range of state-of-the-art baseline methods. Have you considered expanding this comparison to include more recent and advanced approaches? For instance, 'DaViT: Dual Attention Vision Transformers (2022)'. Additionally, for Crowd-AI Collaboration, an insightful comparison could be made with approaches such as the one described in 'Beyond Labels: Empowering Human with Natural Language Explanations through a Novel Active-Learning Architecture (2023)'. Incorporating these comparisons could significantly enrich the context of your study’s evaluation

**Questions:**

There seem to be potential limitations in the scalability and generalizability of the SymLearn framework across various social media platforms and different health crisis contexts. Can you discuss how the framework might be adapted to diverse platforms and crisis situations?

The experimental comparisons and evaluation metrics employed in the study seem to lack comprehensiveness, particularly in the context of potentially imbalanced datasets. Could you provide a rationale for the chosen metrics and discuss the possibility of incorporating more diverse comparison methods in future studies? Specifically, including metrics like AUROC (Area Under the Receiver Operating Characteristic curve) and AUPRC (Area Under the Precision-Recall Curve) could offer more insights, especially for analyzing imbalanced datasets. How might these metrics enhance the evaluation of your framework?

I noticed that the current set of comparative analyses in the study might benefit from a broader range of state-of-the-art baseline methods. Have you considered expanding this comparison to include more recent and advanced approaches? For instance, 'DaViT: Dual Attention Vision Transformers (2022)'. Additionally, for Crowd-AI Collaboration, an insightful comparison could be made with approaches such as the one described in 'Beyond Labels: Empowering Human with Natural Language Explanations through a Novel Active-Learning Architecture (2023)'. Incorporating these comparisons could significantly enrich the context of your study’s evaluation

**Reviewer Confidence:**

3: The reviewer is confident but not certain that the evaluation is correct

**Scope:**

4: The work is relevant to the Web and to the track, and is of broad interest to the community

---

### Official Review · Reviewer_uPuF · 2023-11-22

**Novelty:** 4
**Technical Quality:** 3

**Review:**

This paper proposes a symbiotic human-AI collective learning framework, for Web-based healthcare-policy-adherence assessment (WebHA) problem. Specifically, the aim of this paper is to evaluate people's healthcare policy adherence by utilizing massive amounts of social media image data. This paper designs a novel search space for crowd-AI collaboration and selects a set of optimized WebHA models and crowd workers from this search space through a context-aware multi-armed bandit model. The performance has been demonstrated on two real-world WebHA applications.

Strengths:

S1: An interesting topic for combining AI and human intelligence for healthcare policy adherence assessment.

S2: Development of a novel crowdsourcing interface design that enhances human label quality.

S3: Clear and well-motivated reasoning in the paper.

Weaknesses:

W1: Potential limitations in scalability and generalizability across diverse social media platforms and health crisis contexts.

W2: Insufficient reproducibility.

W3: The experimental comparisons and evaluation metrics are not very comprehensive.

**Questions:**

Q1: For the issues of mask-wearing and adhering to social distancing guidelines studied in this paper, it may be relatively easy for people to assess compliance with mask-wearing and social distancing policies. For other issues, is there a need to provide professional, instructive training and screening for annotators, and would modifying crowdsourcing questions affect the generalizability of this method?

Q2: The performance of this paper is somewhat influenced by the quality of the annotations from crowd workers, posing challenges to reproducibility. Additionally, the codes and datasets are not provided in the manuscript. Therefore, reproducibility is not assured. Please provide the codes and datasets.

Q3: Could you provide more comparisons with more state-of-the-art baselines methods, for example, (1) Deep Neural Network: [Symbolic Discovery of Optimization Algorithms, 2023], [DaViT: Dual Attention Vision Transformers, 2022]（2）Crowd-AI Collaboration: [Beyond Labels: Empowering Human with Natural Language Explanations through a Novel Active-Learning Architecture, 2023]?

Q4: For the imbalancedclassification, it would be more appropriate to additionally provide the area under the precision-recall curve (AUPRC) as metrics in the experiments.

**Reviewer Confidence:**

3: The reviewer is confident but not certain that the evaluation is correct

**Scope:**

4: The work is relevant to the Web and to the track, and is of broad interest to the community

---

### Official Review · Reviewer_JFcu · 2023-11-23

**Novelty:** 5
**Technical Quality:** 4

**Review:**

The article introduces SymLearn, a symbiotic human-AI collective learning framework. It focuses on leveraging the strengths of both AI and human intelligence for Web-based healthcare policy adherence assessment (WebHA), particularly in global health crises like COVID-19 and Monkeypox. The main
contribution is the development of the SymLearn framework, which addresses the limitations of existing human-AI systems by creating a more balanced and effective partnership between human intelligence and AI models. SymLearn comprises two key modules: Symbiotic Crowd-AI Collaboration Space Design (SCSD) and Active Integrated Crowd-AI Co-Learning (AICC). SCSD improves human label quality by guiding them with AI, while AICC uses a context-aware multi-armed bandit model to select optimized combinations of AI models and crowd workers, reducing response delays and improving accuracy.
Strenghts
1. The integration of human and AI intelligence in a symbiotic framework is a novel approach, particularly in addressing complex tasks like Web-based healthcare policy adherence assessment.
2. By leveraging the strengths of both humans and AI, the framework aims to improve the accuracy of AI models while reducing the inherent errors and biases of human judgment.
3. The framework has been tested in real-world scenarios, demonstrating its practicality and effectiveness in actual use cases.
4. The framework’s design suggests potential scalability and adaptability to various domains beyond healthcare, though this would need further exploration.

Weaknesses
1. The use of social media data for healthcare policy assessment raises questions about privacy and ethical considerations, which are not extensively discussed in the paper.

**Questions:**

1. What ethical considerations were taken into account when developing Sym-
Learn, especially concerning the use of social media data? How does the frame-
work ensure privacy and ethical compliance?

**Reviewer Confidence:**

3: The reviewer is confident but not certain that the evaluation is correct

**Scope:**

4: The work is relevant to the Web and to the track, and is of broad interest to the community

---

### Official Review · Reviewer_VRMb · 2023-11-27

**Novelty:** 6
**Technical Quality:** 7

**Review:**

The paper develops a symbiotic human-AI co-learning framework to address a Web-based healthcare-policy-adherence assessment (WebHA) problem. In particular, the authors implement SymLearn, a framework that combines AI models and Human Intelligence (HI), involving crowd workers and crowdsourcing questions to reduce errors in image labels. The framework is empirically validated using two real-world datasets comprising Twitter/X images, each pre-labeled as either adhering or not adhering to social distancing and mask-wearing practices. The results demonstrate that SymLearn surpasses the performance of nine well-established AI models selected for comparison.

Strengths:
1. The paper is well-written, complete, and detailed.
2. The paper is relevant to the conference.
3. It shows a very interesting Human-AI framework, comparing its performances with other AI models.

Weaknesses:
1. A more in-depth discussion regarding the trade-off between performance and costs would enhance the paper by providing a nuanced understanding of the model comparisons. This additional analysis could shed light on the practical implications and feasibility of implementing SymLearn in larger real-world scenario.

**Questions:**

Q1: How might the trade-off between performance and costs impact the practical implementation of the SymLearn framework in real-world scenarios, and what considerations should be taken into account to optimize its feasibility and effectiveness?

**Ethics Review Description:**

/

**Reviewer Confidence:**

1: The reviewer's evaluation is an educated guess

**Scope:**

3: The work is somewhat relevant to the Web and to the track, and is of narrow interest to a sub-community

---

### Decision · Program_Chairs · 2024-01-22

**Decision:**

Accept

**Comment:**

The reviewers appreciate the novelty of the approach, the quality of the writing, and the practical usefulness of the proposed approach.
 Some critical questions were raised, but I don't see any show-stoppers, and the authors provide detailed responses.
 I therefore recommend accepting the paper.